# Translation of Experimental Findings from Animal to Human Biology: Identification of Neuronal Mineralocorticoid and Glucocorticoid Receptors in a Sectioned Main Nerve Trunk of the Leg

**DOI:** 10.3390/cells12131785

**Published:** 2023-07-05

**Authors:** Sascha Tafelski, Jan D. Wandrey, Mohammed Shaqura, Xueqi Hong, Antje Beyer, Michael Schäfer, Shaaban A. Mousa

**Affiliations:** 1Charité–Universitätsmedizin Berlin, Corporate Member of Freie Universität Berlin and Humboldt-Universität zu Berlin, Department of Anesthesiology and Operative Intensive Care Medicine, Campus Charité Mitte and Campus Virchow Klinikum, Charitéplatz 1, 10117 Berlin, Germany; 2Department of Anaesthesiology, Ludwig-Maximilians-University Munich, 80539 Munich, Germany

**Keywords:** mineralocorticoid receptor, glucocorticoid receptor, human peripheral nerve, nociception

## Abstract

The activation of the mineralocorticoid (MR) and glucocorticoid (GR) receptors on peripheral sensory neurons seems to modify pain perception through both direct non-genomic and indirect genomic pathways. These distinct subpopulations of sensory neurons are not known for peripheral human nerves. Therefore, we examined MR and GR on subpopulations of sensory neurons in sectioned human and rat peripheral nerves. Real-time PCR (RT-PCR) and double immunofluorescence confocal analysis of MR and GR with the neuronal markers PGP9.5, neurofilament 200 (NF200), and the potential pain signaling molecules CGRP, Nav1.8, and TRPV1 were performed in human and rat nerve tissue. We evaluated mechanical hyperalgesia after intrathecal administration of GR and MR agonists. We isolated MR- and GR-specific mRNA from human peripheral nerves using RT-PCR. Our double immunofluorescence analysis showed that the majority of GR colocalized with NF200 positive, myelinated, mechanoreceptive A-fibers and, to a lesser extent, with peripheral peptidergic CGRP-immunoreactive sensory nerve fibers in humans and rats. However, the majority of MR colocalized with CGRP in rat as well as human nerve tissue. Importantly, there was an abundant colocalization of MR with the pain signaling molecules TRPV1, CGRP, and Nav1.8 in human as well as rat nerve tissue. The intrathecal application of the GR agonist reduced, and intrathecal administration of an MR agonist increased, mechanical hyperalgesia in rats. Altogether, these findings support a translational approach in mammals that aims to explain the modulation of sensory information through MR and GR activation. Our findings show a significant overlap between humans and rats in MR and GR expression in peripheral sensory neurons.

## 1. Introduction

Mineralocorticoids and glucocorticoids belong to adrenocorticosteroid hormones that play a vital role for controlling various systems in mammals including the metabolic processes, immune system, cardiac system, reproduction, behavior, and cognitive functions. Classically, the glucocorticoids and mineralocorticoids exert their effects through the activation of very closely related glucocorticoid and mineralocorticoid receptors (GR and MR), respectively [1]. Recently, there has been growing attention paid to peripheral neuronal corticoid receptors, since substantial evidence of the expression of mineralocorticoid (MR) and glucocorticoid (GR) receptors in peripheral nerves was referred to inhibitory mechanisms on pain [2]. The latter may inherit potential therapeutic options in different pain conditions, as both genomic [3] and non-genomic [4] pain reduction was described for corticoid agonists and mineralocorticoid antagonists. In recent studies, we were able to show MR and GR expression in the sensory nerve terminals of human and rat skin [2] and, thereby, addressed the lack of translation from non-human animals to patients [5]. Due to considerable overlap between animal and human histology, there is evidence for a common systems approach in mammals that may modulate sensory information by MR and GR activation. This goes along with experimental findings that selective mineralocorticoid receptor antagonists prevent pain development [4,6,7]. Local application of eplerenone, a potent MR antagonist, at the site of an experimentally induced inflammation of dorsal root ganglia prevented the triggered mechanical hyperalgesia and allodynia [6]. Furthermore, canrenoate-K, a different MR antagonist, showed a rapid non-genomic reduction of mechanical hyperalgesia and allodynia in experimental peripheral inflammation [4]. Vice versa, the application of specific MR agonists generated neuronal action potentials and consequently caused mechanical hyperalgesia and allodynia in animal models of pain [4]. Summarizing recent findings, MR inhibition may allow innovative therapeutic options for acute somatic pain [7], chronic neuropathic back pain [6], or chronic inflammatory pain [4]. For the pain relieving effects of glucocorticoid receptor agonists, there is a growing body of data supporting its vital role in acute postoperative pain [8], chronic arthralgia [9], and chronic low back pain [10]. Again, glucocorticoid effects were mainly attributed to long-lasting, genomic mechanisms [11]; however, more recently, fast acting, non-genomic effects were also reported [12]. As GR and MR were identified on peripheral nociceptors in rat and human skin, modulation of neuronal activity is a plausible mechanism in the peripheral nerve system [2]. However, translational studies on GR and MR in the peripheral nerve are still lacking. Early data suggested two corticoid receptors in the central nervous system and described type I (i.e., mineralocorticoid receptor) and type II corticosteroid receptor gene expression [11]. Indeed, until recently, the exact anatomical localization of MR or GR remained unknown with regard to specific subpopulations of peripheral nerve fibers. Immunohistochemical evidence showed GR coexisting with substance P and CGRP in distinct DRG neurons of rats [13]. Precise identification of dorsal root ganglion neurons with GR and MR expression further underscored their role as modulators of nociception [12]. Here, especially in these mixed nerves, expression of pain modulators would be of additional clinical interest: the sciatic nerve with its distal branches of the peroneal and tibial nerve are possible section sites during major limb amputation, and their dissection is most likely associated with the development of neuropathic pain. Therefore, they may serve as potential targets for a local therapeutic intervention [14].

Against this background, we intended to explore the neuronal expression of MR and GR receptors in large mixed nerves in a comparative study on human and animal biology. Therefore, we investigated the local mRNA expression of MR and GR in peripheral nerves, the localization of GR and MR, and their colocalization with pain relevant markers on peripheral mixed nerves. Furthermore, we discussed barriers for histopathologic management and potential differences between human and animal samples, and what functional implications peripheral modulation of MR and GR may have on nociceptive pathways.

## 2. Materials and Methods

### 2.1. Collection of Tissue Samples

Patients scheduled for elective surgical amputation were asked to provide written informed consent for tissue samples for further immunohistochemical analysis. The human sample study adhered to the International Guidelines of Declaration of Helsinki (World Medical Association: http://www.wma.net (accessed on 3 January 2014)) and received approval from the Charité ethical review board (EA1/228/14 and EA4/202/21). Sciatic nerve samples for the animal study were taken from COPCRL rats, following approval by the local animal care committee (G0169/19) and in accordance with the European Directive on animal welfare and care guidelines (2010/63/EU). Tissue samples from peripheral nerve trunks were obtained from two patients undergoing amputation surgery, as well as from three individual rats. These tissue samples were divided into one part, which was quickly frozen and used as a fresh frozen sample for quantitative RT-PCR and into another part that was immediately immersed in fixative solution for double-immunohistochemistry (see more details below). Moreover, for radioligand binding and [^35^S]GTPγS coupling experiments as well as behavioral nociceptive testing, *n* = 5 rats per group were used.

### 2.2. Tissue Preparation

Representative tissue samples of the nerve trunk of the human tibial nerve and sciatic rat nerve were removed (Appendix A) and macroscopically compared in their size and length (Appendix A). It is noticeable that a 2 cm longitudinal section of a peripheral nerve trunk appears much thicker (approximately 5-fold) in humans (presented nerve: average diameter 2.5 mm in humans and 0.5 mm in rats) than in rats.

### 2.3. MR and GR mRNA Detection by RT-PCR

Tissue samples from peripheral nerve trunks obtained from two patients and three different rats were quickly frozen for quantitative RT-PCR. Then, total RNA was extracted from human (*n* = 2) and rat (*n* = 3) peripheral nerve trunks and reverse transcribed into cDNA using an RNeasy (Qiagen, Hilden, Germany) and a cDNA synthesis kit (Takara, Göteborg, Sweden), respectively. The following specific primer pairs for MR (MR homo sapiens, Accession No: NM_000901.4) forward primer, GR (GR homo sapiens, Acession No.: NM_0001762) and 18S (18S homo sapiens Accession No. M10098), were used: MR (forward primer: 5′-AAAGAGCAGTGGAAGGGCAA-3′, reverse primer: 5′-TGAAGTCTGCAAGCAGGACAA-3′), GR (forward primer: 5′-TGCTCCTTCTGCGT TCACAA-3′, reverse primer: 5′-CCATCAGTGAATATCAACTCTGGC-3′), 18S (forward primer: 5′-AAACGGCTACCACATCCAAG-3′, reverse primer: 5′-CCTCCAATGGATC CTCGTTA-3′). Moreover, the following specific primer pairs for MR (Rattus norvegicus Accession No: NM_013131.1), GR (Rattus norvegicus Accession No: NM_012576.2), and 18S (Rattus norvegicus Accession No. NR_046237) were used: MR (forward primer: 5′-CCAAGGTACTTCCAGGATTTAAAAAC-3′, reverse primer: 5′-AACGATGATAGACACATC CAAGAATACT-3′), GR (forward primer: 5′-CATCTTCAGAACAGCAAAATCGA-3′, reverse primer: 5′-AGGTGCTTTGGTCTGTGGGATA-3′), 18S (forward primer: 5′-CGGCTACCACATCCAAGGAA-3′, reverse primer: 5′-GCTGGAATTACCGCGGCT-3′). Their respective dissociation curves are shown in Appendix A. Taqman^®^ Real-Time PCR (Taqman^®^ 7500, Applied Biosystems, Carlsbad, CA, USA) was performed with a SYBR^®^ Green master mix following the manufacturer’s instructions (Applied Biosystems). Amplification was carried out for 40 cycles, each consisting of 15 s at 95 °C for GR and 18S of 30 s at 60 °C. A temperature just below the specific melting temperature (Tm) was employed for detection of fluorescence specific products (GR: Tm 83 °C, 18S: Tm 88 °C). Each tissue sample from humans or rats was measured three times (triplicate). See Appendix A for the list of primers for Taqman RT-PCR.

### 2.4. Immunohistochemistry

All tissue preparations adhered to the protocols described in previous experimental studies [2,12]. They are briefly described as follows: human nerve trunk tissue samples (*n* = 2) were obtained following surgery and immediately fixed in 4% (*w*/*v*) paraformaldehyde in 0.16 M phosphate buffer solution (pH 7.4) for 180 min. Rat nerve trunk tissue samples (*n* = 3) were obtained from naïve COPCRL rats (Janvier Labs) in deep isoflurane anaesthesia. Rats were transcardially perfused with 100 mL warm saline, followed by 300 mL 4% (*w*/*v*) paraformaldehyde in 0.16 M phosphate buffer solution (pH 7.4). After perfusion, the peripheral sciatic nerve was removed, then postfixed in the same fixative for 90 min. Both human and rat tissue were subsequently cryoprotected overnight at 4 °C in PBS containing 10% sucrose. Afterwards, samples were washed with PBS and stored in −80 °C for cryoprotection. Nerve trunk tissue sections (10 mm thick) from both humans and rats were without any pathology and mounted onto gelatin coated slides. To confirm that the selected tissue sections clearly represented neuronal tissue and were without any overt pathology, they were immunostained with the pan-neuronal polyclonal rabbit antibody anti-PGP9.5 and counterstained with the blue fluorescent DNA stain DAPI to identify nuclei of all cellular structures (see Appendix A). Tissue sections were then incubated overnight with the following primary antibodies: polyclonal chicken PGP9.5 (EnCor Biotechnology, Gainesville, FL, USA), polyclonal rabbit anti-GR (private gift from M. Kawata, Koyoto, Japan; this antibody has previously been shown in COS-1 cells with or without GR transfection to be highly specific [12]), and mouse monoclonal antibody against MR (private gift from Prof. Elise Gomez-Sanchez, Jackson, MI, USA), which has previously been shown in heart tissue to be highly specific [15,16,17,18] alone or in combination with guinea pig polyclonal TRPV1 antibody (Neuromics, Minneapolis, MN, USA), monoclonal mouse anti NF200 antibody (Sigma-Aldrich, St. Louis, MO, USA), polyclonal guinea pig CGRP antibody (Peninsula, San Carlos, CA, USA), and polyclonal rabbit anti-Nav1.8 (Sigma). After incubation with primary antibodies, the tissue sections were washed with PBS and then incubated with Alexa Fluor 594 donkey anti-rabbit antibody (Vector Laboratories, Newark, CA, USA) in combination with Alexa Fluor 488 goat anti-guinea pig, or anti-mouse antibody (Invitrogen, Leipzig, Germany). Thereafter, sections were washed with PBS, and the nuclei were stained bright blue with 4′-6-Diamidino-2-phenylindole (DAPI) (0.1 μg/mL in PBS) (Sigma). Finally, the tissues were washed in PBS, mounted in vectashield (Vector Laboratories) and imaged on a confocal laser scanning microscope, LSM510, equipped with an argon laser (458/488/514 nm), a green helium/neon laser (543 nm), and a red helium/neon laser (633 nm; Carl Zeiss, Göttingen, Germany) as described previously [4]. To demonstrate the specificity of staining, the following controls were included as described in our previous studies [4]: omission of either the primary antisera or the secondary antibodies (see Appendix A). Single optical slice images were taken using ×20 Plan-Neofluar air Interface or ×40 Plan-Neofluar oil interface objective lens. See Appendix A for the characterization of used primary antibodies.

### 2.5. In Vivo Administration of Dexamethasone and Aldosterone

GR-agonist dexamethasone (20 µg/20 µL) and MR-agonist aldosterone (40 µg/20 µL) were applied to rats i.th. before nociceptive testing as described in previous experimental studies [3]. Briefly, the needle, through which the catheter was set up, was inserted at a 30° angle between the L5 and L6 vertebras. The catheter (polyethylene PE10 catheter, 15 cm in length, 0.61 mm outer diameter, Portex, Kent, UK) was carefully advanced while rotating it between the thumb and forefinger. Signs of dura penetration were observed by movement of the tail and/or hind limbs. The catheter was then carefully pushed upward 1–2 cm into the i.t. space and the i.t. location of the catheter was confirmed by administration of 10 µL lidocaine 2%, which caused a reversible bilateral hind limb paresis that lasted only 10–15 min [19].

### 2.6. Nociceptive Testing

All rats were subjected to an acclimatization period of a minimum of 72 h (3 days) prior to their use for experimental purposes. For this, rats were placed in a plastic cage for half an hour to acclimatize until cage exploration and major grooming activities ceased. Through a wire mesh at the bottom of the cage, mechanical hind paw withdrawal thresholds were assessed by the application of a calibrated series of von Frey filaments of logarithmic incremental stiffness (Stoelting, Wood Dale, IL, USA) as described previously [12]. Reduction of mechanical hyperalgesia was tested in rats with 4 days of Freund’s complete adjuvant-induced hindpaw inflammation, whereas induction of mechanical hyperalgesia was tested in normal rats.

### 2.7. Maximum Radioligand Membrane Binding Sites and [^35^S]GTPγS Coupling

Membranes were obtained from the lumbar (L3-5) DRG, and [^3^H]Corticosterone and [^35^S]GTPγS binding assays were performed as described previously [12,20]. Saturation binding experiments were performed using GR-agonist [^3^H]Corticosterone (specific activity 40 Ci/mmol, Hartmann Analytic, Braunschweig, Germany). A measure of 50–100 μg of membrane protein was incubated with various concentrations of 1.25–40 nM [^3^H]Corticosterone and 10 μM of the unlabeled GR ligand corticosterone for 1 h at 22 °C in a total volume of 1 mL of binding buffer (50 mM Tris–HCl, 5 mM EDTA, 5 mM MgCl2, 100 mM NaCl, 0.2% bovine serum albumin). All experiments were performed in duplicate and carried out at least five times.

For [^35^S]GTPγS binding assay DRG membranes were incubated in [^35^S]GTPγS (specific Activity 1000 Ci/mmol, Hartmann Analytic, Germany) assay buffer containing 50 mM Tris–HCl, pH 7.4, 5 mM MgCl2, 0.2 mM EGTA, 100 mM NaCl, and 1 mM dithiothreitol. Concentration–effect curves were generated by varying concentrations of dexamethasone (10−12–10−4 M), with 30 μM GDP, and 0.05 nM [^35^S]GTPγS in a total volume of 800 μL. Basal values were obtained in the absence of agonist, and nonspecific binding was measured in the presence of 10 μM unlabeled GTPγS. Bound and free [^35^S]GTPγS were separated by vacuum filtration through GF/B filters and quantified by liquid scintillation counting. All experiments were performed in duplicate and carried out five times.

### 2.8. Statistics

All statistical tests were performed using the Sigma Stat 2.03 software (SPSS Inc., München, Germany). Within the same set of animals, von Frey filament pressure thresholds were measured before and after drug injections, expressed as means ± SD, and statistically evaluated using a repeated ANOVA measurement, followed by a post hoc Dunnett’s test. Saturation binding experiments and [^35^S]GTPγS coupling experiments were analyzed by one-way ANOVA, followed by a post hoc Dunnett’s test. All statistical tests used a two-sided alpha level of <5% and were intended as exploratory.

## 3. Results

### 3.1. Human and Rat Nerve Sample Preparation

Representative tissue samples of the nerve trunk of the human tibial nerve and sciatic rat nerve were removed (Appendix A) and macroscopically compared in their size and length (Appendix A). It is noticeable that a 2 cm longitudinal section of a peripheral nerve trunk appears much thicker (approximately 5-fold) in humans (presented nerve: average diameter 2.5 mm in humans and 0.5 mm in rats) than in rats.

### 3.2. Identification of MR- and GR-Specific mRNA in Peripheral Nerve Trunks of Human and Rat

Using distinct primer pairs for MR and GR in comparison to the internal standard 18S, the expected MR- and GR-specific PCR products were identified in human (*n* = two, triplicate) as well as rat (*n* = three, triplicate) peripheral nerve trunks (Figure 1). The respective cycle threshold (Ct) values revealed that MR (human: 24) and GR (human: 21) values were relatively lower in humans than in rats (MR: 26; GR: 23) indicating a higher mRNA expression. To validate the specificity of our selected primers in human and rat, we show the dissociation curves of the best primer pairs, respectively, chosen from three designed and tested primer pairs each.

### 3.3. Confirmation of Neuronal Structures with the Pan-Neuronal Marker PGP9.5

To confirm that the selected tissue sections clearly represent neuronal tissue and were without any overt pathology, they were immunostained with the pan-neuronal polyclonal rabbit antibody anti-PGP9.5 and counterstained with the blue-fluorescent DNA stain DAPI to identify nuclei of all cellular structures. (Appendix A). The PGP9.5 staining shows abundant nerve fibers, while DAPI blue counterstaining of nuclei identifies other cellular structures.

### 3.4. Detection of Glucocorticoid Receptors in Human and Rat Nerve Trunk

#### 3.4.1. GR and NF200 Colocalization

Double-immunofluorescence confocal microscopy of the rat nerve shows abundant colocalization of GR with NF200. Similarly, there is obvious colocalization of GR with NF200 in the human nerve (Figure 2). Neurofilament 200 is well known to be highly expressed in adult myelinated peripheral axons [21].

#### 3.4.2. GR and CGRP Colocalization

To investigate whether GR localized in the peripheral nerve fibers of sectioned nerve trunks from humans compared to rats, we performed double immunofluorescence staining using the sensory neuronal marker CGRP, which is known to be predominantly expressed in small- to medium-sized unmyelinated C-fiber neurons and thinly myelinated A-δ neurons known to transmit painful stimuli [22,23] (Figure 3).

### 3.5. Detection of Mineralocorticoid Receptors in Human and Rat Nerve Trunks

#### 3.5.1. MR and CGRP Colocalization

In addition to GR, MR is also detectable in the nerve fibers of sectioned nerve trunks from humans and rats (Figure 4). Our cross-sectional picture shows high abundant MR immunoreactivity in human nerve fibers, similarly to the peripheral nerve fibers of rats. Moreover, these MR-immunoreactive neurons colocalize to a high degree with the sensory neuronal marker CGRP in both species.

#### 3.5.2. MR and TRPV1 Colocalization

To better assess the character of these MR-immunoreactive sensory neurons, we costained with a specific antibody for the pain transducing, nonselective cation channel TRPV1 (Figure 5). Our pictures show some colocalization of MR with TRPV1, however, other MR-immunoreactive fibers are not colocalizing with TRPV1 in humans and rats.

#### 3.5.3. MR and Nav1.8

In the present study we identified the Nav1.8 channel in human nerve fibers. For this, we used a specific antibody against the tetrodotoxin (TTX)-resistant voltage-gated Nav1.8 channel, which is known to be specifically expressed in unmyelinated, small-diameter, nociceptive C-fibers (Figure 6). Moreover, double immunofluorescence confocal microscopy showed some colocalization of MR and Nav1.8 in human and rat peripheral nerve fibers, which show some colocalization of MR and Nav1.8. However, some nerve fibers express either MR- or Nav1.8 immunoreactivity alone.

### 3.6. Functional Non-Genomic Effects of GR- and MR-Agonists in Rats

Determining mechanical pressure thresholds with von Frey filament testing showed a significant elevation of mechanical thresholds (reduced hyperalgesia) in rats with hindpaw inflammation (Figure 7A) and a significant decrease of mechanical thresholds (increased hyperalgesia) in the normal hindpaws of rats (Figure 7B). These immediate (within minutes) non-genomic effects are in part explainable by membrane standing steroid receptors, which can be demonstrated by a saturable binding with radiolabeled [^3^H]Corticosterone in membrane fractions of dorsal root ganglia and apparently saturable [^35^S]GTPγS coupling to a putative G protein coupled receptor, following increasing concentrations of the ligand dexamethasone.

## 4. Discussion

In this study, we could show that both MR and GR are expressed in sectioned peripheral rat (sciatic) and human (tibial) nerve trunks (RT-PCR, Figure 1). We could further specify that GR is expressed in peripheral myelinated neurons (NF200 pos., Figure 2) both in rat and human peripheral nerves. GR and MR are also expressed in peripheral unmyelinated and thinly myelinated A-δ neurons (CGRP pos., Figure 3 and Figure 4). MR shows a high degree of colocalization with CGRP in both human and rats, whereas GR colocalizes to some degree in peripheral nerves. Moreover, MR colocalizes both with TRPV1 (Figure 5) and to some degree with Nav1.8 (Figure 6), although in both cases there was a fair amount of nerve fibers expressing MR but not TRPV1 or Nav1.8, respectively. To demonstrate the functional relevance of GR and MR expressed in peripheral sensory neurons, we could show that intrathecal application of a GR agonist immediately reduced mechanical hyperalgesia and intrathecal administration of an MR agonist increased mechanical hyperalgesia in rats. These immediate antinociceptive effects cannot be explained by genomic effects. However, they can be explained by intracellular signaling of the membrane standing steroid receptors, which eventually results in [^35^S]GTPγS coupling to a putative G protein coupled receptor. These findings let us conclude that MR and GR could be future targets for innovative pain treatment options in peripheral nerves.

The results of the PCR showing MR- and GR-specific transcripts, both in human and rat peripheral nerves, confirm our previous findings in human and rat skin tissue [2] and in rat spinal cord and dorsal root ganglia [7,12]. However, the presumably relatively higher mRNA expression of rat versus human neuronal MR and GR might not necessarily be represented by a higher protein expression. Our previous immunofluorescence confocal analysis of human and rat skin revealed strong localization of GR on NF200-positive myelinated mechanoreceptive A-fibers, however, this occurred to a lesser extent with peripheral peptidergic CGRP-positive sensory nerve fibers [2]. Similarly to human and rat skin, here in the nerve fibers of a major human and rat nerve trunk GR also colocalized with NF200 and CGRP, suggesting a possible modulation of nociception. Indeed, GR has been reported to play a role in nociception by prolonging the duration of peripheral nerve blocks via the combination of dexamethasone and local anesthetic solutions [8,24,25]. Due to its significant effect, the use of the GR-agonist methylprednisolone in peripheral nerve block series was also integrated into clinical use in chronic pain conditions [26,27].

The TRPV1, formally known as vanilloid receptor subtype 1 (VR1), is relevant in the detection of noxious thermal and chemical stimuli [28] and, thus, the protection from acute non damaging heat [29]. Its colocalization with MR further underscores its possible role in the modulation of nociception. Capsaicin, a potent activator of TRPV1, is established to treat neuropathic pain by local application of 8% patches [30]. TRPV1 activation induced experimental edema in rabbit skin that was not altered by dexamethasone treatment [31]; data on MR activation on TRPV1-modulated effects in peripheral nerves are currently lacking. Interestingly, experimental application of capsaicin to the sciatic nerve induces a significant depletion of neuropeptides in both the dorsal horn and innervated skin [32]. Moreover, Shaqura et al. demonstrated a distinct genomic regulation of specific pain-signaling molecules, including TRPV1 through MR activation in an animal model of hindpaw inflammation [3].

The Nav1.8 is exclusively expressed on peripheral nociceptive neurons and has been shown to be relevant for NGF-induced thermal hyperalgesia [33], rodent models of osteoarthritis [34], nerve injury and inflammatory pain [35], and especially small-fiber neuropathy [36]. Therefore, Nav1.8 has been the target for the development of pain medication [37,38]. The evidence of the prolonged efficacy of non-selective sodium channel blockers for peripheral nerve blocks with GR agonists [8,24,25] could indicate a use of a combination of specific Nav1.8 blockers with GR agonists.

To demonstrate the functional link between sensory neuron steroid receptors and nociception, we demonstrated, in rats, a reduction in mechanical hyperalgesia following intrathecal application of the GR-agonist dexamethasone and an increase in hyperalgesia following intrathecal administration of the MR-agonist aldosterone. These opposing effects of GR and MR on nociception have been previously described after systemic application [4] and have been supported by the demonstration of aldosterone’s [6] and dexamethasone’s [39] immediate effects on neurons [40]. The demonstrated immediate effects on nociception are only explained by non-genomic effects of MR and GR, which is consistent with previous findings within the central nervous system [41,42]. These non-genomic pathways may be elicited either by directly interfering with intracellular signaling pathways or by interfering with membrane-bound structures such as ion channels and/or G protein coupled receptors [41,42]. Our finding of dexamethasone-stimulated [^35^S]GTPγS coupling to a putative receptor might give a first hint in that direction, as it was previously already speculated that steroids might act on the G protein coupled receptor 30 (GPR30) [42].

For translational trials from rat to human biology, differences in tissue preparation may be important aspects in the interpretation of findings. Since lethal perfusion, as described in animal studies, is not feasible in human sampling, the adaption of established tissue preparation protocols was needed [7,12]. Notably, size differences in peripheral nerves could play a relevant role in tissue preparation, as it is known from external parameters like temperature and may lead to tissue shrinking and vesicular blebbing [43]. As the small molecule formaldehyde penetrates quickly into tissues, tissue size may not be the critical issue for larger nerves to reach an equilibrium, but tissue fixation with protein cross-linking may require long time exposure. With our protocol for tissue preparation, we observed no relevant difference in quality of tissue samples from human and rat.

## 5. Conclusions

In conclusion, this study provides evidence of GR and MR expression in rat and human peripheral nerves. It may serve as anatomical evidence of the expression of MR and GR in different populations of peripheral nerves and as a potential target for the regulation of peripheral sensory nerves. Overall, our findings support a common systems approach in mammals that regulates sensory information through MR and GR activation. There seems to be a significant overlap in humans and rats with regard to MR and GR expression in peripheral nerves. Targeting MR and GR in peripheral nerves may provide a potential for therapy in a number of pain syndromes from a translational standpoint. Future research might provide insight on the critical function corticoid receptors play on peripheral neurons and their influence on nociception.

## Figures and Tables

**Figure 1 cells-12-01785-f001:**
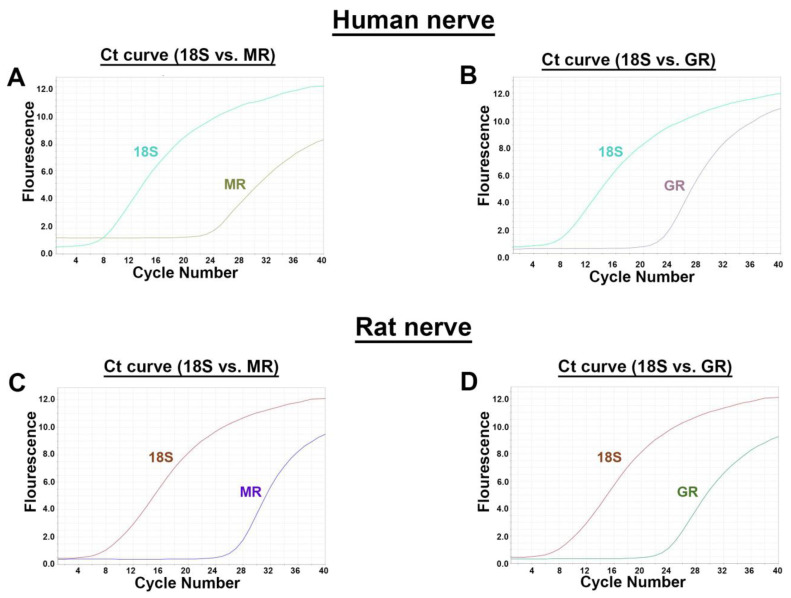
RT-PCR detection of MR and GR mRNA (**A**) in human and rat peripheral nerve trunk. Identification of both MR (**A**,**C**) and GR (**B**,**D**) mRNA by specific primers in human (**A**,**B**) and rat (**C**,**D**) peripheral nerve. For all four combinations, amplification plots with cycle threshold (Ct) values for MR (human: 24; rat: 26) and for GR (human: 21; rat: 23) vs. 18S (human: seven; rat: eight) are provided.

**Figure 2 cells-12-01785-f002:**
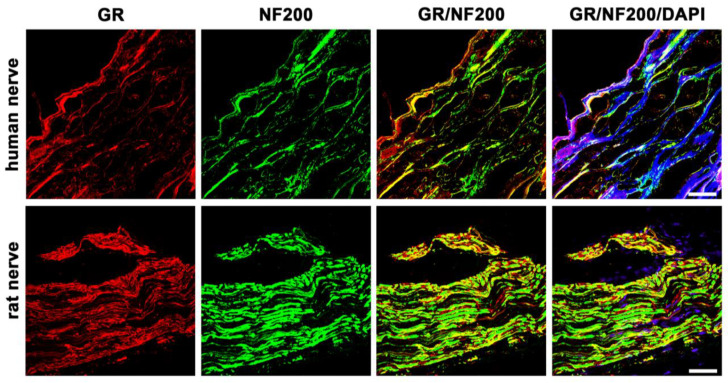
Double immunofluorescent staining for the glucocorticoid receptor (GR) and the neurofilament 200 (NF200, a marker for myelinated neurons) of peripheral nerve trunks from human and rat. Peripheral nerve tissue sections were stained with a rabbit polyclonal anti-GR antibody (red fluorescence) and a monoclonal mouse anti-NF200 antibody (green fluorescence). The pictures show that GR is detectable in the nerve trunk of human and rat and that it colocalizes to a large extent with peripheral myelinated neurons. Nuclear staining of all cells was done with DAPI (a blue-fluorescent DNA stain). Bar = 20 μm.

**Figure 3 cells-12-01785-f003:**
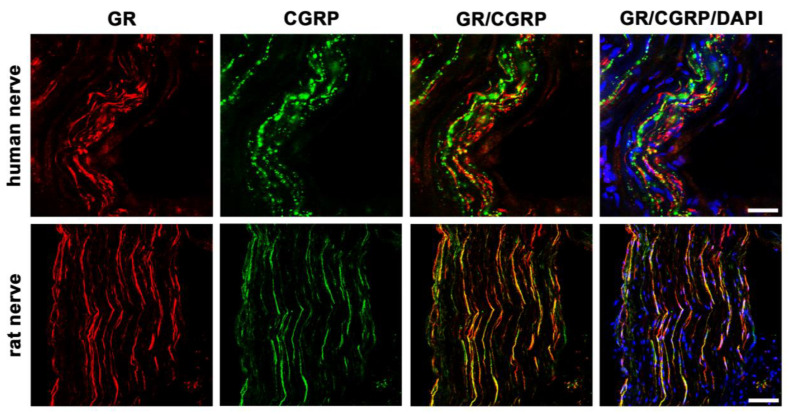
Confocal microscopy of double immunofluorescence of the glucocorticoid receptor (GR) and the sensory neuronal marker CGRP in peripheral nerve trunks of human and rat. Peripheral nerve tissue sections were stained with a rabbit polyclonal anti-GR antibody (red fluorescence) and a polyclonal guinea pig anti-CGRP antibody (green fluorescence). The pictures show that GR is detectable in the nerve trunk of human and rat and that it colocalizes to some extent with the sensory neuron marker CGRP. Nuclear staining of all cells was done with DAPI (a blue-fluorescent DNA stain). Bar = 20 μm.

**Figure 4 cells-12-01785-f004:**
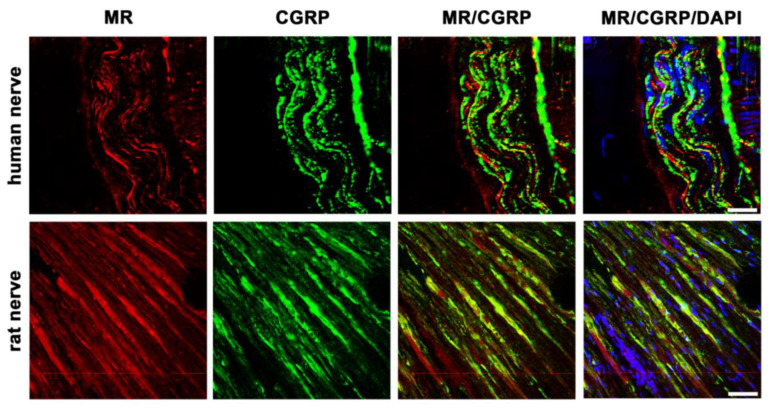
Confocal microscopy of double immunofluorescence of the mineralocorticoid receptor (MR) and the sensory neuronal marker CGRP in peripheral nerve trunks of human and rat. Peripheral nerve tissue sections were stained with a mouse monoclonal anti-MR antibody (red fluorescence) and a polyclonal guinea pig anti-CGRP antibody (green fluorescence). The pictures show that MR is detectable in the nerve trunk of human and rat and that it strongly colocalizes with the sensory neuron marker CGRP. Nuclear staining of all cells was done with DAPI (a blue-fluorescent DNA stain). Bar = 20 μm.

**Figure 5 cells-12-01785-f005:**
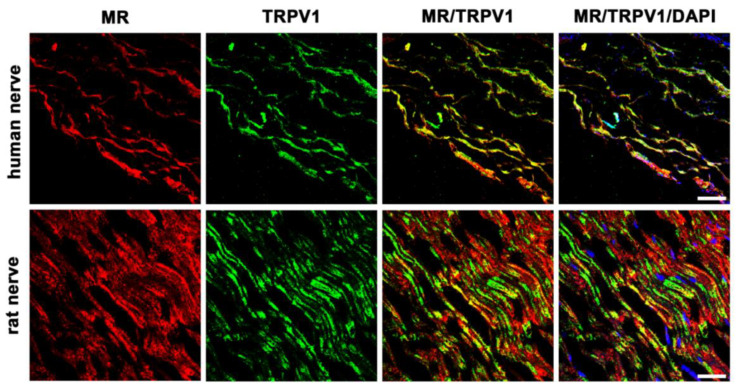
Double immunofluorescent staining for the mineralocorticoid receptor (MR) and the pain transducing/nociceptive nonselective cation channel TRPV1 of peripheral nerve trunks from human and rat. Peripheral nerve tissue sections were stained with a mouse monoclonal anti-MR antibody (red fluorescence) and a polyclonal guinea pig anti-TRPV1 antibody (green fluorescence). The pictures show that MR is detectable in the nerve trunk of human and rat and that it strongly colocalizes with the pain transducing nonselective cation channel TRPV1. Nuclear staining of all cells was done with DAPI (a blue-fluorescent DNA stain). Bar = 20 μm.

**Figure 6 cells-12-01785-f006:**
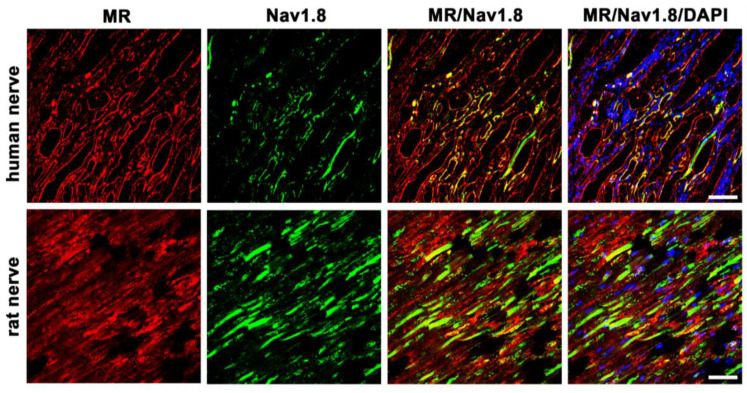
Confocal microscopy of double immunofluorescence of the mineralocorticoid receptor (MR) and the peripheral nociceptive neuron-specific, voltage-gated sodium channel Nav1.8 of peripheral nerve trunks from human and rat. Peripheral nerve tissue sections were stained with a monoclonal mouse anti-MR antibody (red fluorescence) and a polyclonal rabbit anti-Nav1.8 antibody (green fluorescence). The pictures show that MR is detectable in the nerve trunk of human and rat and that it colocalizes with the voltage-gated sodium channel Nav1.8, which is exclusively expressed in peripheral nociceptive neurons. Nuclear staining of all cells was done with DAPI (a blue-fluorescent DNA stain). Bar = 20 μm.

**Figure 7 cells-12-01785-f007:**
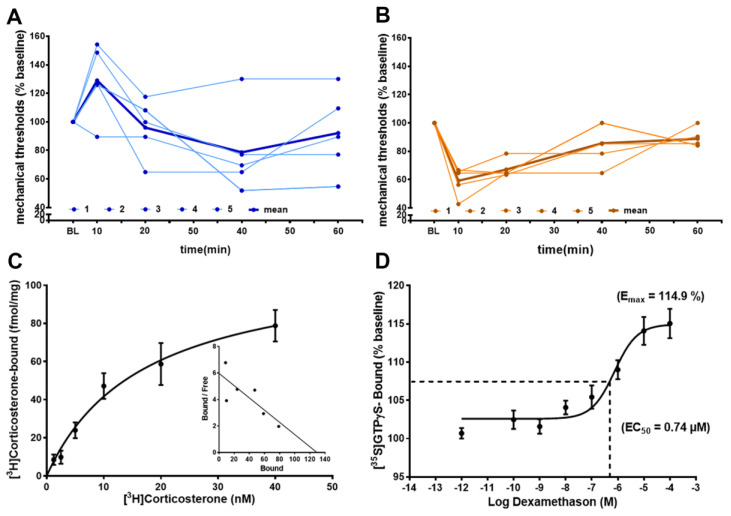
Intrathecal GR-agonist dexamethason reduces hyperalgesia, whereas MR-agonist aldosterone increases hyperalgesia via immediate non-genomic effects. (**A**,**B**) In rats (*n* = five per group) i.th. dexamethason (20 µg/20 µL) reduced, within minutes, hindpaw inflammation-induced hyperalgesia in rats (*p* < 0.05, repeated measurement ANOVA); in contrast, i.th. aldosterone (40 µg/20 µL) evoked, within minutes, mechanical hyperalgesia in normal rats (*p* < 0.05, repeated measurement ANOVA, post-hoc Dunnett’s test). Individual time courses of mechanical von Frey filament thresholds are shown in light blue (**A**) for dexamethasone and light brown (**B**) for aldosterone, while the time courses of mean values are shown in bold blue and bold brown, respectively. (**C**) Membrane fractions from lumbar (L3–L5) dorsal root ganglia of rats show saturable GR binding of [^3^H]Corticosterone with a Bmax = 112 fmol/mg protein and a Kd = 16.9 nM (*n* = 5, in duplicate) (*p* < 0.05, one-way ANOVA, post-hoc Dunnett’s test). (**D**) The same membrane fractions incubated with increasing doses of the GR-agonist dexamethasone reveal saturable [^35^S]GTPγS binding (*n* = five, in duplicate) suggesting immediate non-genomic effects involving G protein coupling (*p* < 0.05, one-way ANOVA, post-hoc Dunnett’s test).

## Data Availability

Data can be accessed upon request by contacting the first author S.T. via e-mail: sascha.tafelski@charite.de.

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
