# Peer review of "Translation of Experimental Findings from Animal to Human Biology: Identification of Neuronal Mineralocorticoid and Glucocorticoid Receptors in a Sectioned Main Nerve Trunk of the Leg"

_cells, 2023, doi:10.3390/cells12131785_

Round 1
Reviewer 1 Report
Dear authors,
I found your research quite well and I appreciated your aims. This paper is closed in the structure and the objective of one you published four years ago (PMID: 30771315). Nevertheless, despite some qualities of your article, I have some comments and I have a doubt about statistics.
Comment 1: “Tissue from peripheral nerve trunks was obtained from two patients undergoing amputation surgery and from three different rats. One part of the tissue samples was used as a fresh frozen sample for quantitative RT-PCR and the other part was used in fixative solution for double-immunohistochemistry". You put this explanation in the results part. It is not the good place. You have to mention it in the methods part. Indeed, when I read your methods part I was wondering how you can do an RT-PCR after putting tissues in formaldehyde. It is impossible.
Comment 2: "Rats were placed in a plastic cage for half an hour to acclimate until cage exploration and major grooming activities ceased". Did you do adaptation days before? Are you sure that 30 minutes of acclimatisation is enough and will not influence your results?
Comment 3: For all the immunofluorescence it misses a negative control as proof that there is not antibody autofluorescence. Without negative control, we cannot conclude. You can provide picture to the editor in supplementary file. Here we have not supplementary data.
Comment 4: As all of us you used antibodies coming from Mississipi (Gomez-Sanchez family). We all appreciate their work. Unfortunately, sometimes their antibodies are not working in some specifications. Do you have a paper where immunofluorescence for the mouse monoclonal antibody against MR. If yes, please quote.
Comment 5: Are 2 humans and 3 animals enough to do statistics for the RT-PCR? I do not think but I am not statistician. Of course for the human samples, It is complicated. But for the animal samples you can increase. Did you calculate the G-Power before your experiment?
For instance, doing an average with only two samples seems to me an error: "It is noticeable that a 2 cm longitudinal section of a peripheral nerve trunk appears much thicker (approximately 5-fold) in humans (presented nerve: average diameter 2.5 mm in humans and 0.5 mm in rats) than in rats.
Comment 6: "The respective Cycle threshold (Ct) values revealed that MR (human: 24) and GR (human: 21) values were relatively lower in humans than rats (MR: 26; GR. 23) indicating a higher expression". Agree. But it is only for the mRNA level. Not for the protein level. In Figure 7, the signal for the MR is stronger in rats compared to humans. Of course, it misses quantifications. But, it shows that the mRNA and the protein levels might be different. I believe that you have to note it somewhere.
Comment 7: Why you did not perform western blot as you did for your last articles? Is it a question of the quantity of the materials?
Comment 8: Did you conclude that a rat is a good model? If yes, please write it clearly.
You did good work but without negative controls I cannot judge a large part of your results. Please put it in the supplementary data. For the statistics, you probably have to briefly discuss with a statistician to be sure that the number of tissues is enough. It is important for the credibility of the manuscript.
Author Response
General:
We re-organized parts of the manuscript due to changes in the review process to enhance readability of the text.
Reviewers' comments: Reviewer #1:
Comment 1: “Tissue from peripheral nerve trunks was obtained from two patients undergoing amputation surgery and from three different rats. One part of the tissue samples was used as a fresh frozen sample for quantitative RT-PCR and the other part was used in fixative solution for double-immunohistochemistry". You put this explanation in the results part. It is not the good place. You have to mention it in the methods part. Indeed, when I read your methods part I was wondering how you can do an RT-PCR after putting tissues in formaldehyde. It is impossible.
Response: Thank you for your suggestion. We have now removed this sentence from the results section and have added the following sentence under Materials and Methods, section “2.1. Collection of tissue samples”: “Tissue samples from peripheral nerve trunks were obtained from two patients undergoing amputation surgery as well as from three individual rats. These tissue samples were divided into one part which was quickly frozen and used as a fresh frozen sample for quantitative RT-PCR and into another part that was immediately immersed into fixative solution for double-immunohistochemistry (see more details below).” (see page 3, lines 100-104)
Comment 2: "Rats were placed in a plastic cage for half an hour to acclimate until cage exploration and major grooming activities ceased". Did you do adaptation days before? Are you sure that 30 minutes of acclimatisation is enough and will not influence your results?”
Response: Thank you, this aspect was not described sufficiently. Indeed, rats were adapted to the testing situation at least three days prior to testing. Therefore, this information has now been included in the methods section: “All rats were subjected to an acclimatisation period of a minimum of 72 hours (3 days) prior to their use for experimental purposes. For this, rats were placed in a plastic cage for half an hour to acclimate until cage exploration and major grooming activities ceased.” (see page 5, lines 205-206)
Comment 3: “For all the immunofluorescence it misses a negative control as proof that there is not antibody autofluorescence. Without negative control, we cannot conclude. You can provide picture to the editor in supplementary file. Here we have not supplementary data.”
Response: We have now added in Suppl. Fig.4 the results of our negative control experiments in which we omitted each individual primary antibody (GR, MR, NF200, CGRP, TRPV1, Nav1.8) and incubated only with the secondary antibody to show that auto-fluorescence is low (see suppl. Figure 4).
Comment 4: “As all of us you used antibodies coming from Mississipi (Gomez-Sanchez family). We all appreciate their work. Unfortunately, sometimes their antibodies are not working in some specifications. Do you have a paper where immunofluorescence for the mouse monoclonal antibody against MR. If yes, please quote.”
Response: We understand the reviewer’s concern that antibody specificity sometimes can be a problem. However, in the present study and extending several previous studies in our laboratory, we used a private MR antibody from Prof. Elise Gomez-Sanchez, Jackson (private gift) after a careful and long process of characterization of many other antibodies including commercial ones. Indeed, the MR antibody provided by Dr. Gomez-Sanchez has been extensively characterized to be specific (Gomez-Sanchez et al., Steroids. 2011; 76: 1541–1545, Gomez-Sanchez et al., Endocrinology, 2006; 147:1343–1348). In addition, we used this MR antibody successfully in several previous studies in human and rat tissue (Tafelski et al., Brain Res. 2019; Mohamed et al., Anesthesiology. 2020; 132: 867-880.; Shaqura et al., J Neuroinflammation. 2020; 17:183; Li et a., Anesthesiology. 2018; 128:796-809). Therefore, we are quite confident about the specificity of the MR-antibody we used which is also represented by the quality of our immunohistochemical pictures. Following the reviewer’s advice, we have now briefly addressed these thoughts in our manuscript and have now added several previous publications using mouse monoclonal antibody against MR from Gomez-Sanchez family. (see page 4 and lines 160-169).
Comment 5: “Are 2 humans and 3 animals enough to do statistics for the RT-PCR? I do not think but I am not statistician. Of course for the human samples, It is complicated. But for the animal samples you can increase. Did you calculate the G-Power before your experiment? For instance, doing an average with only two samples seems to me an error: "It is noticeable that a 2 cm longitudinal section of a peripheral nerve trunk appears much thicker (approximately 5-fold) in humans (presented nerve: average diameter 2.5 mm in humans and 0.5 mm in rats) than in rats.”
Response: For our RT-PCR and immunohistochemical experiments a statistical analysis is not required, since we only wanted to show the existence and putative colocalization of GR, MR, NF200, CGRP, TRPV1, and Nav1.8 but we did not want to make any quantitative presumptions. Since in our first manuscript draft we had not included binding and behavioural experiments, the methods section lacked the information that “(…) for radioligand binding and [35S]GTPgS coupling experiments as well as behavioral nociceptive testing n=5 rats per group were used.” This we have now included in the methods section. (page 3, lines 105-106). Moreover, the appropriate statistical tests for the binding, coupling and behavioural experiments have now been outlined in the 2.8 Statistics section (see page 5, lines 239-241) and in the legend of the respective figures.
Comment 6: "The respective Cycle threshold (Ct) values revealed that MR (human: 24) and GR (human: 21) values were relatively lower in humans than rats (MR: 26; GR. 23) indicating a higher expression". Agree. But it is only for the mRNA level. Not for the protein level. In Figure 7, the signal for the MR is stronger in rats compared to humans. Of course, it misses quantifications. But, it shows that the mRNA and the protein levels might be different. I believe that you have to note it somewhere.”
Response: According to the reviewer’s suggestion, we have now specified “indicating a higher mRNA expression” (page 6, lines 265-266), and have added the following sentence “However, the presumable relatively higher mRNA expression of rat versus human neuronal MR and GR might not necessarily be represented by a higher protein expression.” in the discussion (page 16, lines 428-430) to address the reviewer’s concern.
Comment 7: “Why you did not perform western blot as you did for your last articles? Is it a question of the quantity of the materials?”
Response: Thank you very much for raising this topic. Since the material was limited, we had to prioritize our research questions. Our main focus was to confirm the expression of GR and MR in human nerves in a similar way to rats and not to quantify any differences.
Comment 8: “Did you conclude that a rat is a good model? If yes, please write it clearly.”
Response: This is an interesting question. Regarding tissue modelling, we did not use comparative animal models with different species. To be fair, we cannot conclude from our experiments in this study that rat is a good model. Nevertheless, the rat model is well established in our lab and we have good experiences with it in previous studies (e.g. Tafelski et al., Brain Res. 2019; Mohamed et al., Anesthesiology. 2020; 132: 867-880.; Shaqura et al., J Neuroinflammation. 2020; 17:183; Li et a., Anesthesiology. 2018; 128:796-809).
Comment 9: “You did good work but without negative controls I cannot judge a large part of your results. Please put it in the supplementary data. For the statistics, you probably have to briefly discuss with a statistician to be sure that the number of tissues is enough. It is important for the credibility of the manuscript.”
Response: We have now addressed the reviewer’s concern by adding Suppl. Figure 4 as negative controls for our antibody staining.
Reviewer 2 Report
Abstract section:
1- lines 24.25: “We isolated MR and GR specific transcripts from human peripheral nerve using RT-PCR”.
Please clarify what do you mean by isolating specific transcripts using RT-PCR.
Materials and methods section:
3-lines 114-134: please organise the used antibodies in a table
4- lines 136-158: please add the mentioned primers in a table and calculate the primer efficiency.
5-Line 158: Experiments were done in triplicate, please clarify if you mean technical triplicate.
6-Line 160: 2.5. In vivo administration of dexamethasone and aldosterone, its too short, please explain briefly.
7-Line 164: please describe briefly.
8-Lines 176, 178; please check the name: GR agonist [3H] Corticosterone.
Results section:
9- lines 204-213: please omit from results section, it’s obvious that a human nerve will be larger than a rat nerve. Figures and comment can be presented as a supplementary figure.
10- 3.2 Identification of MR and GR specific mRNA in peripheral nerve trunks of human and rat:
a) Lines 217-218, human (n=2, triplicate), please clarify how do you consider human sample triplicate if u just present an n=2.
b) This analysis is incomplete, presenting the Cycle threshold (Ct) of the house keeping gene and gene of interest are not enough, you have to normalise the expression and present the graph in a 2-ΔΔCT as stated by Livak and Schmittgen; 2001 (doi.org/10.1006/meth.2001.1262).
c) Lines 222-223 three designed and tested primer pairs, please clarify hoe did you test the primer pairs, you have not provided any primer efficiency, primer efficiency must be calculated.
d) Line 225 ,Figure 2 must be removed and substituted with the normalized analysis following the 2-ΔΔCT method .
11- lines 229-242; please omit from results, it will not add any value to confirm the presence of neurons in a tibial and sciatic nerves (unless you are evaluating the regenerated neurons following a nerve trauma). Can be added as supplementary.
12- lines 245- 257: the tissues dimensions are different, it should not be stated that GR localization with NF200 more in rats than in humans without doing a quantification of the colocalization and normalize it to the total area of the tissue, quantification should be presented as a percentage, please perform the quantitative analysis and present a graph with the with colocalization percentage, please perform the quantitative analysis for all the subsequent presented figures.
13- Figure 6: unlike the other presented graphs in figure 6 you present a transverse section of human nerve. Please change to a longitudinal section.
14- lines 325- 334: please rewrite in a more clear way.
15- lines 342- 344 this information should be presented before in the result section.
Discussion section:
16- Please rewrite and insert the new data of quantifications.
please revise the article for English.
Author Response
Abstract section:
1- lines 24.25: “We isolated MR and GR specific transcripts from human peripheral nerve using RT-PCR. Please clarify what do you mean by isolating specific transcripts using RT-PCR.”
Response: Thank you for this question. To avoid misunderstanding, we have replaced the term “transcripts” with “mRNA”. (page 1, line 24)
Materials and methods section:
3-lines 114-134: “please organise the used antibodies in a table.”
Response: According to the reviewer’s suggestion, we have now added a table to show all antibodies used in this manuscript (see supplemental table 1, see page 4 lines 189-190).
4- lines 136-158: “please add the mentioned primers in a table and calculate the primer efficiency.”
Response: According to the reviewer’s suggestion, we have now added a supplemental table to show primers used in this manuscript (see supplemental table 2 see page 3 lines 142-143). However, the primer efficiency can only be determined from established standard curves of the respective target templates which we did not perform because we were not interested in any quantification.
5-Line 158: “Experiments were done in triplicate, please clarify if you mean technical triplicate.”
Response: We have now clarified this misunderstanding by the following sentence: “Each tissue sample of humans or rats was measured three times (triplicate).” (page 3, lines 141-142)
6-Line 160: “2.5. In vivo administration of dexamethasone and aldosterone, its too short, please explain briefly.”
Response: We have now extended our description as follows: “GR-agonist dexamethasone (20 µg/20 µl) and MR-agonist aldosterone (40µg/20µl) were applied to rats i.th. before nociceptive testing as described in previous experimental studies [3]. Briefly, the needle, through which the catheter was set up was inserted in a 30° angle be-tween the L5 and L6 vertebras. The catheter (polyethylene PE10 catheter, 15 cm in length, 0.61 mm outer diameter, Portex, Kent, UK) was carefully advanced while rotating it be-tween the thumb and forefinger. The sign of dura penetration was observed by movement of the tail and/or hind limbs. The catheter was then carefully pushed upward 1–2 cm into the i.t. space and the i.t. location of the catheter was confirmed by administration of 10 µL lidocaine 2% which caused a reversible bilateral hind limb paresis that lasted only 10–15 min.” (page 4-5, lines 193-202)
7-Line 164: “please describe briefly.”
Response: see our changes above.
8-Lines 176, 178: “please check the name: GR agonist [3H] Corticosterone.”
Response: The name is correct but only [3H] is replaced by [3H] (see Shaqura et al., 2016).
Results section:
9- lines 204-213: “please omit from results section, it’s obvious that a human nerve will be larger than a rat nerve. Figures and comment can be presented as a supplemental figure.”
Response: We agree with reviewer and have now presented Figure 2 as Supplemental Figure1.
10- 3.2 Identification of MR and GR specific mRNA in peripheral nerve trunks of human and rat:
- a) Lines 217-218, human (n=2, triplicate), please clarify how do you consider human sample triplicate if u just present an n=2.”
Response: We have now clarified this misunderstanding by the following sentence: “Each tissue sample of humans or rats was measured three times (triplicate).” (page 3, lines 141-142)
- b) “This analysis is incomplete, presenting the Cycle threshold (Ct) of the house keeping gene and gene of interest are not enough, you have to normalise the expression and present the graph in a 2-ΔΔCT as stated by Livak and Schmittgen; 2001 (doi.org/10.1006/meth.2001.1262).”
Response: Since we do not perform any quantification but only confirm the presence of specific mRNA in the present manuscript we do not have to normalise the expression and to present the graph in a 2-ΔΔCT. The comparison of the expression value between rat and human is not our focus in the present manuscript.
- c) “ Lines 222-223 three designed and tested primer pairs, please clarify hoe did you test the primer pairs, you have not provided any primer efficiency, primer efficiency must be calculated.”
Response: We tested the primer pairs by our temperature-dependent dissociation curves of the target specific primer pairs. Each individual dissociation curve represents the primer specific fluorescence change over time with the identification of a single melting temperature at which half of the primer is dissociated from the specific target DNA (see also Suppl. Fig. 2 and its legend).
- d) “Line 225, Figure 2 must be removed and substituted with the normalized analysis following the 2-ΔΔCT method.”
Response: Since we do not perform any quantification but only confirm the presence of specific mRNA in the present manuscript we do not have to normalize the expression and to present the graph in a 2-ΔΔCT. The comparison of the expression value between rat and human is not our focus in the present manuscript.
11- lines 229-242: please omit from results, it will not add any value to confirm the presence of neurons in a tibial and sciatic nerves (unless you are evaluating the regenerated neurons following a nerve trauma). Can be added as supplementary.”
Response: We agree with reviewer and have now presented Figure 3 as Supplemental Figure 3.
12- lines 245- 257: “the tissues dimensions are different, it should not be stated that GR localization with NF200 more in rats than in humans without doing a quantification of the colocalization and normalize it to the total area of the tissue, quantification should be presented as a percentage, please perform the quantitative analysis and present a graph with the with colocalization percentage, please perform the quantitative analysis for all the subsequent presented figures.”
Response: We fully agree with the reviewer and have rephrased/mitigated this paragraph again to avoid misunderstanding regarding the degree of expression of GR with RT200 between rat and human, since this issue is not our focus in the present study: “Double-immunofluorescence confocal microscopy of rat nerve shows abundant col-ocalization of GR with NF200. Similarly, there is obvious colocalization of GR with NF200 in human nerve.” (see page 8, lines 291-295).
13- Figure 6: “unlike the other presented graphs in figure 6 you present a transverse section of human nerve. Please change to a longitudinal section.”
Response: According to the reviewers suggestion, we have now exchanged the transverse section with a longitudinal section (see Figure 4).
14- lines 325- 334: “please rewrite in a more clear way.”
Response: We have now rephrased the sentence as follows: “(…) we performed double immunofluorescence staining using the sensory neuronal marker CGRP which is known to be predominantly expressed in small to medium-sized unmye-linated C-fiber neurons and thinly myelinated A-d neurons which are transmitting painful stimuli.” (see page 9, lines 308-314).
Round 2
Reviewer 1 Report
Dear authors,
I thank you for the reviewing of your manuscript.